# Histone Acyl Code in Precision Oncology: Mechanistic Insights from Dietary and Metabolic Factors

**DOI:** 10.3390/nu16030396

**Published:** 2024-01-30

**Authors:** Sultan Neja, Wan Mohaiza Dashwood, Roderick H. Dashwood, Praveen Rajendran

**Affiliations:** 1Center for Epigenetics & Disease Prevention, Texas A&M Health, Houston, TX 77030, USA; sultanabda@tamu.edu (S.N.); wdashwood@tamu.edu (W.M.D.); 2Department of Translational Medical Sciences, Texas A&M College of Medicine, Houston, TX 77030, USA; 3Antibody & Biopharmaceuticals Core, Texas A&M Health, Houston, TX 77030, USA

**Keywords:** acyl code, cancer, diet, epigenetics, histones, metabolism

## Abstract

Cancer etiology involves complex interactions between genetic and non-genetic factors, with epigenetic mechanisms serving as key regulators at multiple stages of pathogenesis. Poor dietary habits contribute to cancer predisposition by impacting DNA methylation patterns, non-coding RNA expression, and histone epigenetic landscapes. Histone post-translational modifications (PTMs), including acyl marks, act as a molecular code and play a crucial role in translating changes in cellular metabolism into enduring patterns of gene expression. As cancer cells undergo metabolic reprogramming to support rapid growth and proliferation, nuanced roles have emerged for dietary- and metabolism-derived histone acylation changes in cancer progression. Specific types and mechanisms of histone acylation, beyond the standard acetylation marks, shed light on how dietary metabolites reshape the gut microbiome, influencing the dynamics of histone acyl repertoires. Given the reversible nature of histone PTMs, the corresponding acyl readers, writers, and erasers are discussed in this review in the context of cancer prevention and treatment. The evolving ‘acyl code’ provides for improved biomarker assessment and clinical validation in cancer diagnosis and prognosis.

## 1. Introduction

Cancer is a complex disease characterized by the uncontrolled growth and dissemination of aberrant cells, and stands among the foremost contributors to global mortality, resulting in approximately 10 million deaths in 2020 [1]. The development and progression of cancer involves genetic and epigenetic alterations. Epigenetics, the study of heritable changes in gene expression via alterations in DNA methylation, histone modifications, and non-coding RNAs, exerts a crucial role in regulating gene expression and cellular differentiation during development [2]. Aberrant epigenetic changes have been linked to the pathogenesis of several diseases, including cancer [3]. 

Histone PTMs are reversible covalent alterations that affect the structure and accessibility of chromatin, and thereby regulate gene expression. Among these PTMs, histone acetylation and methylation have been extensively reviewed [4], and will not be discussed in detail herein. Recent research has uncovered several new histone modifications, collectively referred to as acyl marks or the ‘acyl code’, which include crotonylation, propionylation, butyrylation, malonylation, succinylation, glutarylation, hydroxybutyrylation, benzoylation, and lactylation (Table 1). Histone acyl marks possess distinct functional characteristics based on their chemical structure, polarity, and reactivity. While the epigenetic ‘writers’ may be shared in some cases, histone acyl marks exhibit preferences for certain ‘readers’ and associated chromatin remodelers [5]. Additionally, the cellular metabolic state can produce distinct acyl-CoA substrates, such as succinyl-CoA or butyryl-CoA. These substrates bind to specific histone regions and regulate gene expression patterns, illustrating the significance of different acyl marks in the interplay between metabolism and epigenetics [6,7,8]. Understanding the roles and mechanisms by which dietary metabolites and metabolism-derived intermediates affect the acyl code and their pathogenic consequences is of paramount importance.

Aberrant histone modifications have been implicated in cancer development and progression. Such modifications occur via various mechanisms, including mutation or deregulation of the enzymes involved in epigenetic control, altered expression levels of the regulatory factors, or via oncogenic substrates produced by cancer metabolism. For example, histone deacetylases (HDACs) are overexpressed in many types of cancer, leading to hypoacetylation of histone and non-histone proteins and repression of tumor suppressor genes [35,36]. Deregulation of chromatin remodelers can also lead to aberrant gene expression and genomic instability, which are among the hallmarks of cancer [37]. In addition to poor dietary habits and associated oncometabolites that increase the risk of cancer [38,39,40,41,42,43], cancer cells undergo metabolic reprograming, triggering epigenetic imbalances to fuel tumor growth and proliferation [44,45,46]. 

It is well established that the digestion and fermentation of dietary fibers by gut microbes, as well as energy metabolism within cells, produce diverse metabolites that can enter cells and generate various acyl-CoAs [47,48]. These acyl-CoAs not only serve as substrates for ATP production but also contribute to PTMs of histone and non-histone proteins [49,50]. These PTMs play vital roles in governing the transcriptome, proteome, and metabolome, thereby exerting significant control over multiple cellular processes [49]. Furthermore, the intricate array of PTMs play a key role in fine-tuning chromatin structure and function. The notion of a ‘histone code’ has gained substantial credibility, linked to PTMs on histone proteins that predominantly regulate DNA transcription [51]. However, the concept of a histone acyl code is relatively new, and it demonstrates that in cancer cells, alterations in metabolic pathways lead to changes in the levels and ratios of acetyl-CoA to acyl-CoA, which, in turn, affect histone acetylation and/or acylation patterns and gene expression. Although some oncometabolites drive tumorigenesis through non-histone protein acylation [16,52,53,54], histone acylation is the predominant epigenetic mechanism. Histone acylation often serves as a marker of chromatin activity, facilitating increased transcriptional output and influencing cellular metabolism through alterations in chromatin structure and function [11,55,56,57,58,59,60,61,62,63]. Additionally, genome-wide analysis has revealed that acyl modifications are associated with gene activation [64]. Notably, the histone acyl code derived from diet and metabolism constitutes an evolving field in epigenetics, offering new insights into the interplay between cellular metabolism and gene regulation [8]. For example, the histone acyl code can influence chromatin remodeling complexes by modifying the structure and function of histones and their interaction with other chromatin-associated proteins. 

No independent acyl writer or eraser has been reported to date. Thus, histone acylation marks are reversibly regulated by the opposing actions of well-documented histone acyltransferases (HATs) and HDACs [65,66]. Just as with acetyl marks, HATs add acyl groups to lysine residues whereas HDACs remove them. The balance between HATs and HDACs is crucial for the proper functioning of cells and tissues, and deregulation of this balance has been linked to various diseases, including cancer [67,68]. Therefore, studying how the intricate repertoire of the acyl code produced from dietary and cellular metabolism is linked to changes in epigenetic regulation could provide new insights into the mechanisms underlying cancer. As epigenetic deregulation is often associated with cancer progression and metastasis [69], understanding the mechanisms and functions of these modifications could lead to the design of new diagnostic and therapeutic strategies. In this regard, histone acylation could provide novel biomarkers for cancer diagnosis, prognosis, and therapy response prediction. Changes in histone acyl code alter chromatin structure and function, leading to changes in gene expression patterns [70]. Understanding this interplay provides insights into metabolic adaptations of cancer cells and their contributions to cancer progression. Targeting acyl readers and enzymes responsible for generating and removing histone acyl marks could provide novel cancer therapies. In this review, we outline current progress in the understanding of histone acyl marks in the realm of cancer biology, as well as the potential therapeutic prospects.

## 2. Histone Acylation

As mentioned above, the histone acyl code encompasses a range of reversible modifications, such as propionylation, butyrylation, crotonylation, succinylation, malonylation, glutarylation, β-hydroxybutyrylation, and benzoylation, among others (Table 1: Overview of histone acyl modifications in cancer). These acylations are dynamic and can be regulated by metabolic changes, providing a diverse repertoire of acyl moieties at any given time. Notably, as previously mentioned, acyl marks are predominantly gene activation marks. They exhibit non-redundancy with histone acetylation [71], and differ in polarity and reactivity, allowing them to differentially regulate gene expression and chromatin structure [50,70]. For instance, specific histone butyrylation, propionylation and β-hydroxybutyrylation marks were associated with the activation of genes involved in lipid metabolism and the response to starvation [11,59], while lysine benzoylation designates promoters of glycerophospholipid metabolism-related genes [62]. In this way, each form of acylation may preferentially recognize specific genomic loci and regulate the expression of distinct sets of genes. While histone acetylation is predominantly recognized by bromodomain readers, acyl marks are often recognized by readers containing DPF and YEATS domains [5]. Histone acylation serves as a distinguishing feature for transcriptional activation during various physiological processes, such as signal-induced gene activation, spermatogenesis, tissue injury, and metabolic stress [8]. Understanding diet- and metabolism-associated acyl code regulation is an evolving new field of precision nutrition [8,72].

The body’s physiological and metabolic state influences precursor molecule availability for the histone acyl code. Factors such as disrupted metabolism of glucose and fatty acids, and the associated regulatory enzymes, play a role in determining the levels of acyl-CoA metabolites [73]. Additionally, rapidly growing cancer cells undergo metabolic alterations that impact acyl-CoA metabolite levels [74]. The availability of acyl-CoA synthases is a crucial factor influencing the diversity and dynamics of histone acylation [75]. These enzymes convert precursor molecules into specific acyl-CoA metabolites, and their levels and activity influence the availability of different acyl-CoA metabolites and subsequently lead to the formation of different types of acylated histones. Additionally, the enzymatic action of HATs and the proclivity of each acyl-CoA metabolite for non-enzymatic acylation are key factors that can impact the diversity and dynamics of histone acylation [76]. 

Histone acetylation is one of the most well-studied PTMs in chromatin. It entails the transfer of an acetyl group from acetyl-CoA to the ε-amino group of lysine residues in histone tails. This modification is catalyzed by HATs, including p300 and lysine acetyltransferase 2A (KAT2A), which exhibit relatively low binding affinity for catalyzing histone crotonylation and succinylation as compared to the canonical acetyl-CoA substrate [58,77]. On the other hand, histone crotonylation is a modification that involves the transfer of a crotonyl group to lysine residues, and is facilitated by p300/CBP-associated factor (PCAF) [58]. Like histone acetylation, histone crotonylation has also been found to be particularly enriched in active gene promoters and enhancers [78]. Nevertheless histone crotonylation and succinylation contribute to chromatin relaxation, rendering DNA more accessible to transcription factors and other regulatory proteins [79,80]. Conversely, histone deacetylation, carried out by HDACs, leads to chromatin compaction and gene repression.

Histone propionylation and butyrylation involve the transfer of propionyl and butyryl groups, respectively, to lysine residues in histone tails. These modifications are mainly catalyzed by sirtuins (SIRTs) and exert similar effects on chromatin structure and gene expression as acetylation [81]. Succinylation, malonylation, and glutarylation involve the transfer of succinyl, malonyl, and glutaryl groups, respectively, to lysine residues on histone tails. As summarized in Table 1: Overview of histone acyl modifications in cancer, these modifications are also mediated by HATs, such as KAT2A (hGCN5), which possesses corresponding acyl-transferase activity. Given the reactivity of acyl-CoA metabolites like succinyl-CoA and malonyl-CoA towards lysine residues, they can also undergo non-enzymatic histone acylation [47]. On the other hand, β-hydroxybutyrylation involves the transfer of a β-hydroxybutyryl group to lysine residues and is carried out by the enzyme p300/CBP, which exhibits β-hydroxybutyrate dehydrogenase activity [82]. This modification is notably enriched in the liver and is believed to play a role in metabolic regulation [82].

### 2.1. Mechanisms of Histone Acylation

Histone acylation involves the interplay between enzymatic activity, metabolic signaling, chromatin structure, and effects on gene expression. For instance, histone acetylation is regulated by the availability of acetyl-CoA, a metabolite produced during cellular metabolism [83,84,85,86]. Acetyl-CoA levels are influenced by metabolic state, nutrient availability, and stress responses [87]. Relative to acetylation, other histone acyl marks are 1–5% as abundant and correlate with cellular levels of the respective acyl-CoA donors [7].

Histone acylation associated HATs and HDACs are subject to regulation by metabolic signaling pathways and other factors [88]. Concomitant to the well-studied HATs and classical family of zinc-dependent HDACs, other enzymes can also regulate histone acylation. For example, SIRT5 relies on NAD^+^ and can remove malonyl and succinyl groups from histone lysine residues through demalonylation and desuccinylation, respectively [89,90]. These modifications are thought to play important roles in regulating metabolism and mitochondrial function. Acetylation, propionylation, and butyrylation can relax chromatin structure, making DNA more accessible for transcription factors and other regulatory proteins [91,92]. Conversely, deacetylation, depropionylation, and debutyrylation can lead to chromatin compaction and gene repression [64,93]. 

Lysine acylation occurs through enzymatic as well as non-enzymatic actions of acyl CoA-thioesters, acyl phosphates, and α-dicarbonyls [94]. Unlike enzymatic acylation, non-enzymatic acyl lysine modifications accumulate in various proteins, particularly in the aging process [95]. The mechanism of other histone acylations, such as crotonylation, succinylation, malonylation, and glutarylation, are less well understood but are thought to play roles in the regulation of gene expression and chromatin structure [60,61], although their precise roles in gene regulation are still being studied. 

### 2.2. Histone Acylation Writers and Erasers

KATs are a diverse group of HAT enzymes that transfer acetyl groups from acetyl-CoA to the ε-amino group of lysine residues present in proteins, including histone tails. Several KAT families, including GNAT (Gcn5-related *N*-acetyltransferase), MYST (Moz, Ybf2/Sas3, Sas2, and Tip60), p300/CBP, and TAF (TATA-binding protein-associated factor) [96] are multi-functional enzymes. They are responsible for transferring various acyl groups, such as acetyl, crotonyl, butyryl, propionyl, malonyl, succinyl, and others, to the lysine residues of histone proteins, with decreased acyl-transferase activity for bulkier acyl-CoAs [97].

Several metabolic enzymes that catalyze the interconversion of acyl substrates contribute to the diversity of the histone acyl code (Figure 1). For instance, malonyl-CoA is produced in the cytosol during the synthesis of fatty acids from citrate. This process involves the enzymatic action of ATP citrate lyase (ACL), which converts citrate into acetyl-CoA, followed by the conversion of acetyl-CoA to malonyl-CoA by acetyl-CoA carboxylase (ACC) [98]. Another enzyme, propionyl-CoA carboxylase (PCC), generates D-methylmalonyl-CoA from propionyl-CoA, which is then transformed into L-methylmalonyl-CoA by methylmalonyl-CoA epimerase (MCEE) [99]. Lastly, methylmalonyl-CoA mutase facilitates the formation of succinyl-CoA [99,100]. Gut microbiome-produced 3-hydroxybutyryl-CoA dehydrogenase (crotonase) converts betahydroxybutyryl-coA into crotonyl-CoA [101], highlighting the role of the microbiota in regulating the diversity of acyl-CoA substrates. 

The HDACs that remove acetyl/acyl groups from histone and non-histone proteins include four classes. Class I consists of Rpd3-like proteins and includes HDAC1, HDAC2, HDAC3, and HDAC8. Class II, Hda1-like proteins, includes HDAC4, HDAC5, HDAC6, HDAC7, HDAC9, and HDAC10. Class III, NAD^+^-dependent Sir2-like proteins, comprises SIRT1, SIRT2, SIRT3, SIRT4, SIRT5, SIRT6, and SIRT7. Lastly, there is a single enzyme in the Class IV protein, which is HDAC11 [102]. These HDACs exhibit differences in subcellular localization, function, and regulatory mechanisms, allowing them to participate in diverse cellular processes and contribute to gene expression regulation, chromatin remodeling, and other important cellular functions. The specificity or preference of an HDAC for acetyl versus other histone acyl marks has not been adequately addressed in the current literature. 

### 2.3. Histone Acylation Readers

Histone modifications are read and interpreted by a diverse array of proteins called ‘readers’, which recognize and bind to specific PTMs. Based on the structural domains, the main families of readers are bromodomain-containing proteins (BRDs), chromodomain-containing proteins (CRDs), tudor domain-containing proteins, PHD finger-containing proteins, and YEATS domain-containing proteins [103,104]. These epigenetic readers play diverse essential cellular functions. They can directly modify histone marks or serve as effector proteins, influencing the functional consequences of histone modifications by translating the histone code into actionable changes. Readers can also recognize and bind to specific epigenetic marks, thereby enabling the recruitment of molecular machinery to modify chromatin structure [105,106].

Recent studies revealed preferential reader selection between histone acetyl and acyl marks [103]. BRDs and CRD-containing proteins typically recognize acetylated and methylated lysine residues on histones, respectively [107]. However, DPF and YEATS domain-containing readers preferentially recognize longer acyl forms of lysine residues, such as crotonylation, butyrylation, and propionylation [103]. The binding specificity of these readers relies on their affinity in binding to aromatic acyl groups [18,30]. 

As noted above, currently there are no specific writers or erasers exclusively dedicated to ‘non-acetyl’ acylation. Therefore, recruitment of histone acylation readers becomes a crucial determinant of chromatin accessibility in modulating the expression of specific genes. Interestingly, under various metabolic, developmental, or disease-related conditions, YEATS and DPF domain-containing readers interacted with larger acyl groups, remaining bound to chromatin, while BRD-containing proteins were excluded [103]. One of the DPF-domain-containing proteins, DPF2, serves as an accessory component of the BAF-family chromatin remodeler has been reported to exert a repressive role in myeloid differentiation [108]. 

## 3. Dietary Metabolites Regulating Histone Acylation

Evidence has accrued for diet-associated bioactive compounds and intermediary metabolites affecting histone acylation marks, including the following: Butyrate, propionate, and acetate are short-chain fatty acids (SCFAs) produced by gut microbiome-mediated fermentation of dietary fiber. These metabolites inhibit HDAC activity and increase histone acetylation status [109];Polyphenols found in various fruits, vegetables, and beverages, including resveratrol and curcumin, act on SIRTs and other HDACs to alter histone acetylation status and gene expression [110,111,112,113];Omega-3 fatty acids abundant in fatty fish, flaxseeds, and walnuts, have been implicated in regulating histone acetylation [114]. They influence the activity of HATs and HDACs, promoting a favorable balance between histone acetylation and deacetylation [115,116];Vitamin B3 (niacin) is involved in energy metabolism as a precursor for the coenzyme nicotinamide adenine dinucleotide (NAD^+^), which is required by SIRTs for deacetylase activity. By affecting NAD^+^ availability, niacin can indirectly modulate histone acylation [117];Glucose utilization, microbiota-derived SCFAs, or dietary fat metabolism can impact acetyl/acyl-CoA ratios, thereby affecting overall histone acetylation patterns [118,119,120,121]. Since most histone acylation competes for the same HATs, the acetyl/acyl-CoA ratios in different cellular pools dictate which acylation pattern occurs on histones [118,121];Dietary antioxidants such as vitamins C and E, and certain polyphenols, modulate cellular redox status and signaling pathways involved in histone acetylation [122]. Additionally, nutrient-sensing pathways, such as the mammalian target of rapamycin (mTOR) pathway, can integrate dietary and metabolic signals to influence histone acylation [123]. Among the nutrient-sensing signaling pathways that govern histone PTMs, the sucrose non-fermenting/AMP-activated protein kinase (AMPK/Snf1) and carbohydrate response element binding protein (ChREBP) pathways play pivotal roles. For instance, AMPK/Snf1 acts as a histone kinase [124], not only phosphorylating but also regulating the activity of several HATs and HDACs through enzyme phosphorylation [125]. Moreover, this pathway influences histone acetylation and deacetylation by controlling levels of acetyl CoA and NAD^+^ levels [125].

### 3.1. The Role of Dietary and Metabolism-Derived Histone Acylation in Cancer Development and Progression 

Many cancers are influenced by non-genetic/environmental risk factors. For instance, tobacco products, tanning beds, UV exposure, alcohol consumption, toxin exposure, and poor dietary habits have been reported to increase the risks of lung cancer, skin cancer, liver cancer, and colorectal cancer, respectively [126,127]. Non-genetic risk factors often cause epigenetic derangements that underlie cancer development. Once oncogenesis is established, cancer cells are also known for their metabolic reprogramming and adaptability, which enable survival and proliferation within the tumor microenvironment [128,129], for which altered metabolism and subsequent epigenetic deregulation play roles [130,131]. In this section of the review, we explore how the metabolic rewiring of cancer cells contributes to the development and progression of tumors by altering histone acylation.

The mechanisms by which poor dietary habits influence histone acylation patterns and cancer risk are complex (Figure 1). Briefly, lifestyle and dietary habit driven alterations in nutrient availability and oncogenic-driven metabolic reprogramming within cancer cells influence the levels of crucial metabolites that govern signaling pathways and epigenetic processes [132]. Consequently, an altered metabolic state and changes in acyl modifications lead to the recruitment of epigenetic writers, chromatin remodelers, erasers, and readers, resulting in a distinct histone acylation landscape that connects cellular metabolism to the epigenome [133]. 

Deficiencies in essential nutrients like folate [134,135], vitamin B12 [136], and iron [137], which function as critical cellular substrates and cofactors, can impair the function of enzymes involved in histone acylation, leading to abnormal histone modifications and potentially promoting oncogenesis. Altered metabolite levels and imbalanced nutrient utilization can also affect the activity of HATs and HDACs, which regulate histone acylation. Cancer cells, in contrast to normal cells, exhibit elevated methionine cycle activity and rely on external or dietary methionine for sustained growth [138]. The significance of methionine metabolism in cancer biology is linked to its role in GSH biosynthesis, Polyamine Synthesis, and as a donor of methyl groups for DNA and histone modification [138]. Additionally, there is an increasing risk of cancer susceptibility due to exposure to carcinogens such as NOCs, PhIP, PAHs, and HAAs, which can result from the consumption of thermally processed meat [139], as well as other carcinogens associated with smoking [140]. Poor dietary habits, including regular consumption of highly processed foods and insufficient fruits and vegetables, can promote chronic inflammation and oxidative stress in the body [141], which can induce changes in histone acylation marks leading to deregulated cellular processes promoting oncogenesis [142]. 

Metabolic reprogramming is a critical factor driving cancer progression, supporting energy generation, the biosynthesis of anabolic molecules [143], and maintaining the optimal cellular redox states within cancer cells [144]. Solid tumors often exhibit the Warburg effect and hypoxia, contributing to cancer cell reprogramming [145]. Unlike normal cells, cancer cells rely on aerobic glycolysis, the pentose phosphate pathway, the hexosamine pathway, and the serine biosynthesis pathway. This increased glycolytic activity can overwhelm mitochondria, leading to the production of reactive oxygen species [45]. Lactate dehydrogenase plays a pivotal role by converting pyruvate to lactate, preventing mitochondrial import, and maintaining NAD^+^ homeostasis [146]. Lactate, with roles in reversing the Warburg effect and modifying histones [63,147], holds significant importance. Cancer cells also rapidly consume glutamine, utilizing it as a nitrogen donor and carbon source for anabolic pathways [148]. Oncogenic signals further drive metabolic reprogramming by enhancing glucose and glutamine transporters [149,150] and modulating metabolic enzyme activity [151]. These altered metabolic pathways are vital conduits, supplying the necessary metabolic intermediates and cofactors for epigenetic modifiers. Consequently, cancer metabolism, marked by significant changes in cellular metabolite levels compared to normal conditions, intricately intertwines with cancer epigenetics [45,152]. For example, the acetylation of histone H3 lysine 9 (H3K9ac) and histone H3 lysine 27 (H3K27ac) have been shown to be regulated by the activity of the acetyl-CoA synthetase enzyme (ACSS2), which catalyzes the conversion of acetate to acetyl-CoA [153]. ACSS2 is upregulated in various cancers, and increased levels of H3K9ac and H3K27ac have been observed in cancer cells [154,155]. These modifications are associated with increased expression of oncogenes. For example in colorectal cancer, an enzyme called ACL, responsible for converting citrate to acetyl-CoA, is suppressed, leading to a decreased nuclear acetyl-CoA reservoir [83]. During glucose deprivation, cancer cells also utilize glutamine as a substrate for the production of acetyl-CoA in the tricarboxylic acid (TCA) cycle, which, in turn, increases histone acetylation to support the proliferation and growth of tumor cells [156,157,158]. Similarly, propionylation and butyrylation of histones have also been implicated in cancer development. The enzymes responsible for these modifications are regulated by metabolic pathways involved in the breakdown of fatty acids [47]. Dysregulation of these pathways leads to altered histone propionylation and butyrylation patterns, which can affect gene expression and contribute to tumorigenesis.

The cross-talk between diet, microbes and host cells also influences cancer outcomes [159,160]. For example, green leafy vegetables such as spinach can alter gut microbes [161], potentially resulting in the generation of microbial enzymes and metabolites serving as molecular messengers. Microbial metabolites can also alter the tumor microenvironment, comprising of a variety of cell types and inflammatory mediators, thereby influencing epigenetic events that play a role in cancer progression and the effectiveness of immunotherapy [160,162]. One of the most common fermentation products of gut microbiome is SCFAs and its metabolites that inhibits HDACs, exerting an epigenetically mediated anti-cancer function [163,164,165,166].

Histone acylation can also change chromatin structure by modulating the interaction between histones and other chromatin-associated proteins. For example, the acetylation of H3K56 has been shown to enhance the binding of the chromatin assembly factor CAF-1, which is involved in nucleosome formation [167]. Other studies have demonstrated that the H4K5 acylation/acetylation ratio fine-tunes BRD4–chromatin interactions highlighting the balance between histone acetylation and acylation [77]. This balance, regulated by metabolic processes, may serve as a widespread mechanism that governs the functional genomic distribution of bromodomain factors [168]. Thus, the recruitment of chromatin-remodeling complexes, associated readers, and eraser complexes that, in turn alter chromatin structure and gene expression, relies on metabolic dynamics within the tumor microenvironment [103,169].

Protein acylation also plays a significant role in shaping the immunosuppressive tumor microenvironment by influencing immune cell exhaustion, activation, and infiltration [170]. Through its regulation of immune cell activation, infiltration, and antigen presentation, protein acylation can influence the formation of an immunosuppressive tumor microenvironment [170]. Cancer cells also utilize histone lactylation as a mediator of immunosuppression [63,171]. Lactate increases histone lactylation and leads to heightened expression of Arg1 and other genes that mediate the transition toward the immunosuppressive M2 macrophage phenotype, thereby restraining immune cell activity in the tumor microenvironment [63,171,172]. Epigenetic rewiring that is intimately connected to cancer metabolism could be one of the mechanisms of cancer immune escape.

### 3.2. Metabolism-Derived Histone Acyl Codes as Cancer Biomarkers

Global histone hypoacetylation is a biomarker of cancer etiology [173,174], and the deregulation of metabolic pathways can lead to alterations in histone acylation patterns that contribute to cancer development and progression (Table 2). Such changes can function as biomarkers for diagnosis and prognosis, as exhibited for breast, prostate, and colorectal cancer [173,175]. For instance, histone H3K9ac and histone H3K27ac, which are considered active histone mark for normal cells, are aberrantly elevated in prostate cancer [175]. ACSS2 is often upregulated in various cancers [154,155] along with altered expression of HATs, HDACs and associated epigenetic reader proteins. For instance, the binding of acetylation reader ENL to H3K9ac and H3K27ac has been observed in acute myeloid leukemia and is associated with increased expression of oncogenes that can be used as biomarkers for diagnosis or prognosis [176]. In another study, increased acetylation of H2BK120, H3.3K18, and H4K77 in liver cancer tissues were biomarkers of unfavorable prognosis. In an independent clinical cohort of hepatocellular carcinoma (HCC) patients, these markers correlated with decreased survival rates and increased recurrence rates [177].

Changes in other histone acylation marks, such as propionylation and butyrylation, were identified in cancer cells and could be promising biomarkers. For example, the propionylation of histone H3K23 in U937 leukemia cells surpass those in non-leukemia cells by at least six-fold. Furthermore, a significant drop in propionylation levels occurred during monocyte differentiation of U937 cells, suggesting that the initial hyperpropionylation in U937 cells might serve as a specific marker of leukemia development [186].

The levels of certain metabolites, such as 2-hydroxyglutarate (2HG), were associated with histone acylation patterns and may serve as potential biomarkers for cancer diagnosis and prognosis [187]. Increased levels of 2HG have been observed in several types of cancer, and this metabolite has been shown to inhibit the activity of histone demethylases, leading to altered histone methylation and gene expression [187,188,189].

Gene mutation in adenomatous polyposis coli (APC) may dictate the onset of colorectal cancer (CRC), but recent epidemiological studies have shown that the majority of young adults diagnosed with CRC do not possess hereditary syndromes or germline mutations typically associated with CRC [190,191]. Remarkably, the conventional clinical criteria used to identify individuals at higher risk of CRC often prove inadequate in these cases [192,193], indicating the need for epigenetic-based biomarkers for screening specific cancers.

## 4. Targeting Histone Acylation for Cancer Prevention and Therapy

Cancer prevention involves proactive measures to reduce cancer risk through lifestyle choices, avoidance of carcinogens, and possibly utilizing medications or vaccines. Environmental and lifestyle factors, encompassing radiation, toxins, pollutants, infectious agents, and diet, influence epigenetic events [194,195]. Disruption of these events leads to abnormal gene expression, notably contributing to severe diseases like cancer. Fortunately unlike genetic mutations, epigenetic changes are potentially reversible, offering a crucial avenue for cancer prevention and therapy [196]. The anticancer role of dietary bioactive compounds and phytochemicals mediated by histone PTMs has been previously reviewed [197,198,199,200,201]. Several clinical trials involving natural products and diet interventions for cancer therapy have also been extensively reviewed [202,203,204,205,206]. This section explores how dietary and metabolism-derived histone acylation events can be used as an attractive target for cancer prevention and therapy.

### 4.1. Targeting Histone Acylation for Cancer Prevention

It is well documented that dietary and lifestyle factors can affect the metabolism-derived histone acyl code and modify cancer risk. For example, certain dietary compounds such as butanoates [161], which are produced by gut microbiota from dietary fiber, can promote histone acetylation and reduce cancer risk [164]. Similarly, exercise and physical activity can affect the metabolism of fatty acids and improve histone acylation patterns [156,207,208]. Several bioactive compounds from the diet have been reported to play a role in preventing cancer through epigenetic mechanisms [209,210] (Figure 1). Hence, reversing the impact of aberrant histone acylation is one approach to preventing the early development and progression of cancer.

Essential nutrients like folate, vitamin B-12, selenium, and zinc, alongside dietary compounds such as sulforaphane, tea polyphenols, curcumin, and allyl sulfur compounds, are part of an expanding arsenal that influences epigenetic processes [210,211] by targeting enzymes involved in histone acylation, such as HATs and HDACs [164,212]. Emerging evidence also suggests that metabolism-derived histone acylations may be involved in regulating gene expression in response to nutrient availability, oxidative stress, and other environmental cues [8,11,213]. For example, the levels of histone acetylation, butyrylation, and succinylation have been shown to change in response to caloric restriction, fasting, or high-fat diets [59,213]. Additionally, targeting metabolic pathways that produce acyl-CoA metabolites, such as fatty acid metabolism, can also be addressed with dietary or pharmacological agents to modify histone acylation patterns and reduce cancer risk. Specifically, fatty acid synthesis inhibitors such as soraphen A, cerulenin, orlistat, TOFA, GSK165, and UB006 have demonstrated antitumor efficacy in cancers such as neuroblastoma [214], prostate cancer [215], and colorectal cancer [216,217,218]. Although targeting histone acylation has been suggested to potentially prevent or slow down cancer development and progression [8,170], additional investigation is warranted to comprehensively elucidate the mechanisms that underlie the connection between altered histone acylation and cancer. This research is crucial for developing effective interventions that can be used in clinical settings.

### 4.2. Targeting Histone Acylation for Cancer Therapy

Histone acylation has been implicated in the pathogenesis of various diseases, including cancer (Table 2). These marks will be discussed in brief.

#### 4.2.1. Targeting Acylation Writer and Eraser

Numerous HDAC inhibitors have undergone development and evaluation in both preclinical and clinical investigations for treating cancer, inflammatory diseases, and metabolic disorders [219,220]. These inhibitors have demonstrated the ability to trigger apoptosis and inhibit tumor growth across diverse cancer types such as lymphoma, leukemia, breast cancer, prostate cancer, and lung cancer [221]. The mode of action of HDAC inhibitors entails suppressing HDAC activity, elevating histone acylation levels and yielding anticancer effects, as outlined in several reviews [221,222,223]. Gut microbiota butyrate production also triggers HDAC inhibition, leading to elevated expression of IFN-γ and granzyme B in cytotoxic T cells (CTLs) [165]. New mechanistic insight was reported for HDAC inhibition by linoleate and butyrate metabolites acting via the IFN-γ pathway to mediate reactivation of immune related genes for antitumor response in a preclinical model of CRC [164]. The working hypothesis was that epigenetic suppression of MHC cell surface presentation could be rectified by correctly positioning neoepitopes to engage the host immune system at the adenoma stage. Exploring the types of histone acylation change required to restore functional MHC complexes on the surface of cancer cells warrants further investigation.

Dietary HDAC inhibitors have also demonstrated anticancer effects linked to histone acetylation status while minimizing the likelihood of adverse effects [224], including sulforaphane [225], polyphenols [226], and spinach metabolites [164]. Research has pinpointed other histone acylation marks as potential therapeutic targets. For instance, the levels of histone succinylation were elevated in certain types of cancer [14], and inhibition of the desuccinylase SIRT5 reduced tumor growth in preclinical models [227,228]. Similarly, histone butyrylation levels linked to insulin resistance and diabetes were addressed by inhibiting the butyryltransferase CBP, improving glucose homeostasis in mouse models. In vivo, gut microbiota-derived butyrate inhibited class I HDACs, thereby affecting histone decrotonylation in mice colons [21]. In vitro studies also indicated that HDAC3 possessed decrotonylase activity [229], suggesting a potential target for HDAC3-specific inhibitors.

Targeting HDACs can also work for nonhistone protein acylation, like palmitoylation. Specifically, the palmitoylation of interferon gamma receptor 1 (IFNGR1) alters its protein–protein interactions. Instead of associating with optineurin, palmitoylated IFNGR1 binds to the adaptor protein complex 3 subunit delta-1 (AP3D1), leading to lysosomal degradation of IFNGR1. This degradation hinders the IFNγ and MHC-I pathways, contributing to immune evasion [230]. Conversely, depalmitoylation of IFNGR1 promotes its stability and the functionality of downstream MHC-I signaling, crucial for effective antigen presentation to T cells [170,230]. Additionally, HDAC2 inhibits PD-L1 acetylation, enhancing nuclear localization and immune checkpoint activation [231]. Meanwhile, P300 promotes MEF2D acetylation, boosting PD-L1 transcription [232]. PD-L1 palmitoylation, facilitated by ZDHHC3 and ZDHHC9, prevents lysosomal degradation, contributing to T cell exhaustion [233,234]. Furthermore, PCAF and GCN5-mediated acetylation enhances Rae-1 stability, activating NK/T cell killing ability. Conversely, P300-driven TRIB3 acetylation hinders T cell infiltration by dampening CXCL10 transcription. SIRT1-mediated deacetylation of p53 promotes TAM infiltration through CXCL12 secretion [235]. Leveraging the immune system through epigenetic drug intervention holds promise for both cancer prevention and therapy.

In addition to targeting epigenetic erasers, other inhibitors of histone acyl-modifying enzymes are also being developed as potential cancer therapies. For example, inhibitors of the histone acetyltransferase (HAT) writer CBP/p300 suppress tumor growth in preclinical models of breast and lung cancer [236]. Consequently, there is a growing interest in CBP/p300 inhibitors and protein degraders as promising therapeutic agents for cancer treatment, with the potential for translation into clinical settings [237]. In the case of breast and prostate cancer, CBP/p300 regulate nuclear hormone receptor signaling [238]. Targeting CBP/p300 may be tissue specific and context dependent, adding to the paradoxical roles in tumor suppression and oncogene actions [238,239].

#### 4.2.2. Targeting Acylation Readers

Recent studies reported improved antitumor outcomes through epigenetic combination therapy via HDAC plus acetyl reader inhibition [240,241,242]. Small-molecule BET inhibitors, such as JQ1, have entered clinical trials [243,244]. Tea and soy polyphenols have been shown to inhibit the non-BET family member BRD9, triggering DNA damage and apoptosis in colon cancer cells [245]. Human MOZ (KAT6A) and DPF2 (BAF45d) use their double PHD finger domains to bind various histone lysine acylations, favoring Kcr, followed by Kpr and Ku [18,30,103,246]. The existence of distinct acylation readers with a preference for specific histone modifications [18,103,246] underscores the significance of these approaches in future investigations of histone acylation reader-targeted therapeutics.

### 4.3. Current Approaches and Future Directions in Targeting Histone Acylation for Cancer Interception

Despite the significant challenges, identifying and addressing the critical hurdles outlined above holds the promise for effective new acyl code-based anticancer therapies [247]. Developing site-specific and more precise molecular tools for targeted acylation or deacylation to control the expression of anticancer therapy genes remain an aspirational pursuit [248,249].

Presently, HDAC inhibitors stand out as the most extensively studied compounds targeting histone acyl-modifications. Several HDAC inhibitors, including vorinostat, romidepsin, and belinostat, have been approved by the FDA for the treatment of various types of cancer [250]. Other approaches include the development of HAT inhibitors and other histone acyl-modifying enzymes such as butyryltransferases, crotonyltransferases, and propionyltransferases [247]. Inhibitors of the HAT p300/CBP showed promise in breast and lung cancer models [236]. Additionally, inhibitors for butyryltransferase GCN5 and the crotonyltransferase PCAF demonstrated antitumor effects in preclinical models [237,251]. Although this review focuses only on histone acylation, there are also non-histone protein acylation and acyltransferases linked to cancer. For instance, a homologous recombination (HR) protein MRE11 is lactylated by CBP in response to DNA damage. High lactate levels in cancer cells lead to MRE11 lactylation and chemoresistance, providing insights into the role of cellular metabolism in DSB repair and chemotherapeutic response [252]. A succinyl transferase OXT1-mediated succinylation of beta-lactamase-like protein (LACTB) inhibits its proteolytic activity, leading to HCC progression [253] indicating potential use of OXCT1 inhibitors for such cancers [254].

The diversity and complexity of histone acyl modifications and their biological functions are areas under intense exploration. Innovative drug delivery methods, such as nanotechnology-based approaches, show promise in improving the bioavailability and efficacy of histone acyl-modifying enzyme inhibitors [255]. Addressing possible toxicity concerns and resistance mechanisms might necessitate combination strategies with immune-based therapies [256,257,258]. Finally, understanding the pharmacokinetic properties of histone acyl-modifying enzyme inhibitors, such as bioavailability and metabolism, is needed for optimizing efficacy and safety in vivo.

### 4.4. Challenges in Targeting Histone Acylation

Translating histone acylation-based treatments faces several challenges, most notably the requirement for extensive proteomic-based screening of histone and non-histone protein acylation in bodily fluids like peripheral blood, fecal samples, or saliva. Such screening methodologies aim to identify early diagnostic markers or therapeutic targets for various diseases, including cancer [259]. Moreover, the heterogeneous nature of cancer cells demands the identification of specific histone acyl codes tailored to various cancer types and individual patient profiles. Ensuring specificity in targeting these modifications is complicated by the overlapping substrate specificity and activity of modifying enzymes, and a consideration of non-enzymatic lysine acylation on non-histone proteins [67,260]. Additionally, delivery of therapeutic agents to cancer cells within the tumor microenvironment is complex, while minimizing off-target effects [260]. Furthermore, the rapid adaptability of cancer cells to changes in their metabolic milieu can result in resistance to therapies targeting histone acylation [261]. Concerns regarding toxicity, including hematological and cardiac adverse effects, along with the influence of pharmacokinetic properties on efficacy and safety in vivo, further complicates the development and clinical use of histone acyl-modifying enzyme inhibitors for cancer therapy.

## 5. Conclusions

Diet- and metabolism-derived histone acylation marks have been implicated in cancer epigenetics, but their relative contributions to overall disease pathogenesis remain underexplored. Complex and dynamic change in histone modifications, catalyzed by specific enzymes, influence gene expression, chromatin structure, and cellular behavior. Understanding the significance and role of dietary and metabolism-derived histone acyl code in cancer epigenetics has the potential to unveil new cancer biomarkers and therapeutic targets. While targeting histone acylation holds promise for cancer therapy, challenges such as the lack of inhibitors for the specific enzymes need to be addressed. Future research endeavors should focus on unraveling the mechanisms of diet and metabolism-derived histone acylation changes, aiming to develop more effective cancer therapies and precision immunoprevention.

## Figures and Tables

**Figure 1 nutrients-16-00396-f001:**
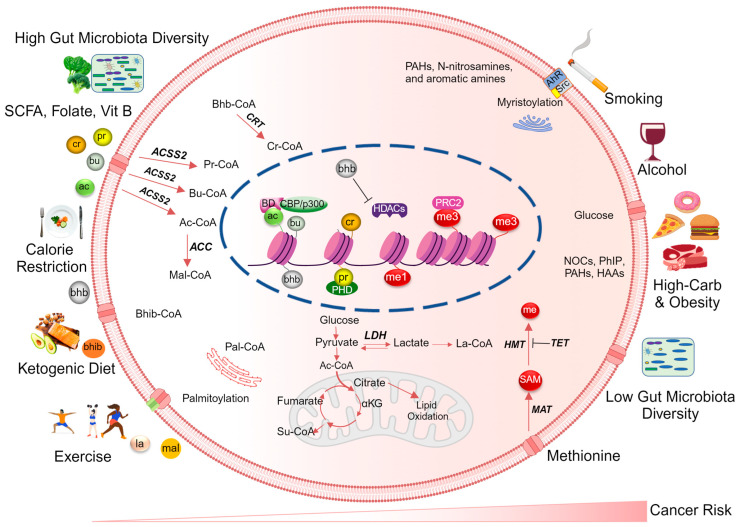
Dietary metabolites regulate cancer risk by modulating the histone acylation landscape. Dietary and metabolic alterations represent non-genetic/environmental risk factors influencing histone acylation during cancer development and progression. The right side of the diagram illustrates risk factors that increase cancer development through epigenetic mechanisms, such as an increase in histone methylation and a low acetylation/acylation ratio. On the left side, the diagram demonstrates how a healthy lifestyle and dietary metabolites that increase the diversity of the gut microbiota alter histone acetylation/acylation patterns. Increased levels of histone acyl-CoA precursors from both dietary and cellular metabolism contribute to chromatin decondensation through HDAC inhibition and the removal of repressive histone marks, thereby reducing the risk of cancer. Key abbreviations: MAT: methionine adenosyltransferase, HMT: histone methyltransferase, TET: ten-eleven translocation, ACC: acetyl-CoA carboxylase, ACSS2: acetyl-CoA synthetase 2, BD: bromodomain reader, PHD: plant homeodomain reader, HDACs: histone deacetylases, SAM: S-adenosyl methionine, PRC2: polycomb repressive complex 2, αKG: alpha ketoglutarate, PAHs: polycyclic aromatic hydrocarbons, HAAs: heterocyclic aromatic amines, NOCs: *N*-nitroso compounds, PhIP: 2-amino-1-methyl-6-phenylimidazo[4,5-b]pyridine, me3: H3K27me3, me1: H3K4me1, AhR: aryl hydrocarbon receptor, CRT: crotonase, me: methyl. Created with BioRender.com, accessed on 26 January 2024.

**Table 1 nutrients-16-00396-t001:** Overview of histone acyl modifications in cancer.

Type of Acylation	Chemical Nature	Dietary/Metabolic Source	Writers	Readers	Erasers	References
Acetylation (Ac)	Hydrophobic	CHO & SCFA from gut microbes, glycolysis, TCA	p300/CBP, HAT, GNATs	BRD3, BRD4, PBRM1	All HDAC family	[8,9,10]
Propionylation (Pr)	Hydrophobic	SCFA from dietary fiber & gut microbes, TCA	p300/CBP, GNATs, MYSTs	YEATS, DPF	SIRT1,2,3	[11,12,13,14,15]
Butyrylation (Bu)	Hydrophobic	SCFA from dietary fiber & gut microbes, TCA	p300/CBP, GNATs, HBO1	YEATS, DPF	SIRT1,2,3	[9,15,16,17]
Crotonylation (Cr)	Hydrophobic	SCFA from dietary fiber & gut microbes, TCA	p300/CBP	YEATS, DPF	SIRT1,2,3, HDAC3	[9,18,19,20,21]
Benzoylation (Bz)	Hydrophobic	N/A	HBO1	YEATS, DPF		[5,9,22]
β-Hydroxybutyrylation (Bhb)	Polar	Ketogenic diet, starvation,	p300/CBP	YEATS, DPF	SIRT3, HDAC1,2,3	[23,24,25]
2-Hydroxyisobutyrylation (Bhib)	Polar	SCFA, Amino acid metabolism	P300, MYSTs	YEATS, DPF	N/A	[16,26]
Lactylation (La)	Acidic	Glycolysis, lactate from exercise, LGSH	p300	N/A	HDAC1,3	[16]
Malonylation (Mal)	Acidic	Citrate metabolism, FAO	N/A	N/A	SIRT2,5	[27,28]
Succinylation (Succ)	Acidic	TCA	p300/CBP, GNATs, CPT1A, GCN5	YEATS	SIRT5, 7	[17,29,30]
Glutarylation (Glu)	Acidic	TCA, amino acid metabolism	p300, GCN5	N/A	N/A	[9,26]
O-GlcNacylation (GlcNac)	Polar	Pentose–phosphate pathway	N/A	N/A	N/A	[31]
Palmitoylation (Pal)	Hydrophilic	Edible oils, HFD	LPCAT1	N/A	APT, PPT SIRT6	[32,33]
Myristoylation (Myr)	Hydrophilic	Edible oils, HFD	N/A	N/A	SIRT2, 6	[33,34]

APT: acyl protein thioesterases, PPT: palmitoyl–protein thioesterase, CHO: carbohydrate, TCA: tricarboxylic acid, SCFA: short-chain fatty acid, GNATs: Gcn5-related *N*-acetyltransferases, FAO: fatty acid oxidation, HFD: high-fat diet, LGSH: lactoylglutathione, LPCAT1: lysophosphatidylcholine acyltransferase I, CPT1A: carnitine acyltransferase I, MYST: lysine acetyltransferases (Moz, Ybf2/Sas3, Sas2, and Tip60), HBO1: histone acetyltransferase binding to ORC1, N/A: data not available.

**Table 2 nutrients-16-00396-t002:** Aberrant histone acylations linked to cancer.

Histone Acylation Type	Cancer Type	Association with Cancer	References
Global H3K18ac, H3K9ac, H3K12ac	Prostate	Elevated levels correlate with prostate cancer risk	[175]
Global losses of H3K16ac	Leukemia, lymphoma, breast, colorectal, lung, prostate, cervical	A hallmark of human tumor cells	[173]
H3K23pr	Medulloblastoma, leukemia, glioma, colorectal	Low H3K23pr contributes to cancer development	[178]
Global histone Kcr	Esophageal, colon, pancreatic, lung	Low Kcr is associated with cancer	[179]
	HCC	Kcr levels correlate with HCC progression	[180]
	Prostate	Kcr levels correlate with prostate cancer malignancy	[181]
H3K9bhb	HCC	High H3K9bhb correlates with HCC progression	[182]
Global Khib	Pancreatic	Khib is a tumor promoter in pancreatic cancer	[183]
H3K18la	Melanoma	High H3K18la enhances melanoma	[184]
H3K9la and H3K56la	HCC	High H3K9la and H3K56la increase the proliferation and migration of liver cancer stem cells	[185]
H3K79succ, H3K122succ	Glioblastoma	High H3K79succ promotes the proliferation and development of glioma cells	[77]
Global histone Kbz	HCC	Kbz is involved in HCC progression	[22]

H3K18ac: histone H3 lysine 18 acetylation, H3K9ac: histone H3 lysine 9 acetylation, H3K12ac: histone H3 lysine 12 acetylation, H3K16ac: histone H3 lysine 16 acetylation, H3K23pr: histone H3 lysine 23 propionylation, histone Kcr: global histone lysine crotonylation, HCC: hepatocellular carcinoma, H3K9bhb: histone H3 lysine 9 β-hydroxybutyrylation, Khib: global histone lysine 2-hydroxyisobutyrylation, H3K18la: histone H3 lysine 18 lactylation, H3K9la: histone H3 lysine 9 lactylation, H3K56la: histone H3 lysine 56 lactylation, H3K79succ: histone H3 lysine 79 succinylation, H3K122succ: histone H3 lysine 122 succinylation, Kbz: global histone lysine benzoylation.

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
