# Peer review of "Histone Acyl Code in Precision Oncology: Mechanistic Insights from Dietary and Metabolic Factors"

_nutrients, 2024, doi:10.3390/nu16030396_

Round 1
Reviewer 1 Report
Comments and Suggestions for Authors
This is a very interesting review and timing is perfect for updating the comprehensive advance and knowledge of histone acyl code beyond histone acetylation and methylation as well as corresponding acyl readers, writers, and erasers of histone post-translational modifications (PTMs) in precision oncology. The authors also summarized the role of dietary and metabolic factors in regulating histone acyl code during tumor progression. Some concerns need to be addressed before publication.
Major concerns:
1. Page 10: “For example, inhibitors of fatty acid synthesis have been shown to reduce histone acylation and cancer risk in preclinical models [192, 193]. Although targeting histone acylation has been suggested to potentially prevent or slow down cancer development and progression [8, 158]”. Please clarify specific inhibitors of fatty acid synthesis and cancer types.
2. Page 11: “Targeting HDACs can also work for nonhistone protein acylation, like palmitoylation, which plays a crucial role in antigen presentation, where the interaction between Optineurin (OPTN) and Adaptor Related Protein Complex 3 Subunit Delta 1 (AP3D1) impedes the recognition of Interferon Gamma Receptor 1 (IFNGR1)”. I don't quite understand what the description of OPTN and AP3D1 has to do with palmitoylation.
3. All the elements in Figures should have their corresponding descriptions in the manuscript. But the metabolism of methionine as shown Figure 1 is not mentioned in the manuscript. What NOCs, PhIP, PAHs, HAAs mean? Why dose La-CoA appear twice? Pr PTM recognized by PHD? Therefore, Figure 1 requires carful preparation.
4. Recently, Ma et al. found that OXCT1, an enzyme catalyzing ketone body oxidation, functions as a specific lysine succinyltransferase to contribute to tumor progression (doi.org/10.1016/j.molcel.2023.11.042). It will be better to add this novel specific succinylation writer OXCT1 in Table 1.
5. Page 9, paragraph 1: Chen et al. found that MRE11 is lactylated by CBP in response to DNA damage, their findings unveil lactylation as a key regulator of HR, providing fresh insights into the ways in which cellular metabolism is linked to DSB repair (doi.org/10.1016/j.cell.2023.11.022). It will be better to discuss this paper in this review.
6. It will better to summarize the effects of these histone PTMs on gene expression in Table 1.
7. Page 12: in my opinion, the order of 4.3 and 4.4 should be reversed. The authors can decide whether to change or not.
Minor concerns:
Page 4, line 6: “lead”, not “leads”
Page 4, paragraph 3 mentioned that “Histone propionylation and butyrylation are catalyzed by HDACs”, but SIRT1/2/3 were summarized in Table 1. Which one is right?
Page 4, paragraph 3: p300/CBP (CREB-binding protein); please delete “(CREB-binding protein)” in page 5, line 9
Page 5: Whether there is an extra space after “(Figure 1)”?
Page 5, paragraph 6, line 2: “CD-containing proteins” refers to “CRDs” or other proteins?
Page 6, line 21: “they” or “that”?
Page 6, lines 23/24 (Page 8, line 9): NAD+
Page 7: Figure 1 and Figure 1 legends: “ACSS2”, not “ACCS2”
Page 7: Figure 1 legends: “right side”, “left-side” (The writing style is not uniform)
Page 7: Figure 1 legends: Whether there is an extra space after “AhR”?
Page 8: Line 24: “acetyl-CoA”, not “Acetyl-CoA”
Page 11: paragraph 3: Whether there is an extra space between “marks” and “as”?
Author Response
We appreciate the opportunity to respond to the comments made by Reviewer 1 on our manuscript entitled "Histone Acyl Code in Precision Oncology: Mechanistic Insights from Dietary and Metabolic Factors." We believe the feedback has significantly enhanced the quality of our paper, and we have addressed each point as follows:
- Specificity of Fatty Acid Synthesis Inhibitors: Clarifications have been provided by specifying inhibitors such as Soraphen A, Cerulenin, Orlistat, TOFA, GSK165, and UB006, along with their impacts on various cancer types, as shown on page 10.
- Role of OPTN and AP3D1 in IFNGR1 Palmitoylation: The explanation has been refined on page 12 to emphasize the effects of IFNGR1 palmitoylation on protein stability and immune evasion.
- Refining Figure 1: We have revised Figure 1 and provided an explanation on page 7 about methionine metabolism and the role of carcinogens like NOCs, PhIP, PAHs, and HAAs. The recognition of histone propionylation by PHD domain readers is now cited with reference [103].
- Inclusion of Ma et al.’s and Chen et al.’s Findings: We have acknowledged the significance of OXCT1 in 'non-histone' protein acylation, as mentioned now in Section 4.3 on page 13. We have also incorporated the implications of MRE11's lactylation in the same section. Additionally, Table 1 has been updated to summarize the different aspects of 'histone' acylation, which is a focus of the review article.
- Histone PTMs' Effects on Gene Expression in Table 1: We have emphasized that acyl modifications are predominantly associated with gene activation on page 3.
All minor concerns, including typographical errors and requests for clarification, have been addressed to enhance clarity and accuracy.
Reviewer 2 Report
Comments and Suggestions for Authors
The manuscript entitled Histone Acyl Code in Precision Oncology: Mechanistic Insights from Dietary and Metabolic Factors is an interesting and scientifically valuable review. The authors point out that histone post-translational modifications (PTMs), including acyl tags, act as a molecular code and play a key role in translating changes in cellular metabolism into lasting patterns of gene expression. Diet and metabolism-derived histone acylation marks have been implicated in cancer epigenetics. Complex and dynamic change in histone modifications, catalyzed by specific enzymes, influence gene expression, chromatin structure, and cellular behavior. Specific types and mechanisms of histone acylation, shed light on how dietary metabolites reshape the gut microbiome, influencing the dynamics of histone acyl repertoires. Although targeting histone acylation is promising in cancer therapy, a significant problem is the lack of inhibitors of specific enzymes. The manuscript is divided into subsections supported by well-selected literature. In vivo and in vitro should be written in italics on page 11. Figure 1 is very interesting. Contains 234 items of cited publications. The manuscript may help develop more effective cancer therapies.
Author Response
We appreciate the opportunity to respond to the comments made by Reviewer 2 on our manuscript entitled "Histone Acyl Code in Precision Oncology: Mechanistic Insights from Dietary and Metabolic Factors." We thank Reviewer 2 for their positive feedback. The manuscript has been corrected to italicize "in vivo" and "in vitro" on page 11, as suggested.
Reviewer 3 Report
Comments and Suggestions for Authors
This manuscript is an intriguing novel review paper focusing on the relationship between diet and the histone Acyl-code. The manuscript, overall, is well-structured and well-written. I have a few minor suggestions that could enhance its readability and depth.
1. It would be beneficial to incorporate information about the types of cancer in Table 1.
2. Could there be any dietary specificity corresponding to the acyl code mentioned in Table 2?
3. It is appropriate to include the status of anticancer clinical trials utilizing specific diets, particularly the ketogenic diet.
4. A discussion on the connection between nutrient metabolism signaling and the histone code, citing examples such as AMPK, mTORC, ChREBP, and SCD1, would be a valuable addition.
Author Response
We appreciate the opportunity to respond to the comments made by Reviewer 3 on our manuscript entitled "Histone Acyl Code in Precision Oncology: Mechanistic Insights from Dietary and Metabolic Factors." We believe the feedback has significantly enhanced the quality of our paper, and we have addressed each point as follows:
- Types of Cancer in Table 1: We have summarized the effects of histone acylations linked to specific cancers in Table 2, while Table 1 provides a general overview of different histone acylations.
- Dietary Specificity for the Acyl Code in Table 2: This aspect is addressed in Table 1. Due to limited literature on cancer-specific dietary sources, this detail is not included in Table 2.
- Anticancer Clinical Trials and Diets: On page 11, we have elaborated on clinical trials, including references that previously reviewed the clinical aspects of dietary bioactives, phytochemicals, and dietary interventions such as ketogenic diets.
- Connection Between Nutrient Metabolism Signaling and the Histone Code: On page 6, we have added a discussion on the AMPK and ChREBP pathways and their roles in histone PTM.
- Order of Subsections 4.3 and 4.4: We have rearranged these sections as suggested by the reviewer to ensure a more logical flow of information.